# DR4/DQ2 haplotype confers susceptibility to T1DM with early clinical disease onset: A retrospective analysis in a tertiary-care hospital in Italy

**Silvia Ricci**[1,2]*, **Francesca Perugia**[3], **Barbara Piccini**[4], **Lorenzo Lodi**[1,2], **Francesco Pegoraro**[1], **Mattia Giovannini**[1,2], **Giovanni Rombolà**[5], **Giancarlo Perferi**[1], **Sonia Toni**[4], **Chiara Azzari**[1,2]

1 Section of Pediatrics, Meyer Children's Hospital, Florence, Italy, 2 Department of Health Sciences, University of Florence, Florence, Italy, 3 Department of Translational Research and the New Technologies in Medicine and Surgery, University of Pisa, Pisa, Italy, 4 Diabetology Unit, Meyer University Children's Hospital, Florence, Italy, 5 Genetics Diagnostics—Laboratory of Immunogenetics and Transplant Biology, Careggi Hospital, Florence, Italy

* silvia.ricci@meyer.it

**Data Availability Statement:** All relevant data are within the paper and its Supporting Information files.

## Abstract

### Introduction

T1DM is the most frequent form of diabetes in children. It has a multifactorial pathogenesis in which genetic, environmental and immunological factors are involved. Among genetic explanations a major role is attributed to second class HLA genes, with the greatest risk associated with the simultaneous presence of the haplotypes DR3DQ2 and DR4DQ8. Based on results obtained in other countries, the aim of this research is to verify a possible association between the haplotype DRB1 * 04: 05-DQA1 * 03-DQB1 * 02 and the onset of T1DM among Italian children with possible genotype-phenotype correlations. Greater knowledge of genes which increase or decrease susceptibility is important for genome analysis.

### Materials and methods

165 patients with type 1 diabetes treated at the Diabetology Unit of the Meyer Children's University Hospital, were clinically analyzed. Data relating to age at diagnosis, pancreatic anti-beta cell autoimmunity, comorbidities with date of diagnosis and family history were retrospectively collected from medical data. A case-control study was conducted to investigate the HLA types of the patients compared to a control group of 819 Tuscan donors enrolled in the National Bone Marrow Donor Register. Typing was carried out using the Eurospital "DIABEGEN" kit, currently in use at the immunology laboratory of the Meyer Children's University Hospital.

### Results

Mean age at diagnosis was 9.3 years; most children (97%) had anti-pancreatic beta cell autoimmunity; the anti-insulin antibody (IAA) was more frequent among children with early

**Funding:** he author(s) received no specific funding for this work.

**Competing interests:** The authors have declared that no competing interests exist.

clinical disease onset (0–5 years of age). From the case control comparison performed on HLA typing, it emerged that the greatest risk for the development of type 1 diabetes is conferred by the haplotypes DR3DQ2 and DR4DQ8, but in addition to these haplotypes, already known in other countries, we identified another haplotype, DR4DQ2 (DRB1 * 04: 05-DQA1 * 03-DQB1 * 02) which appears to predispose children to type 1 diabetes (p value 2.80E-08) and it is associated with early clinical disease onset (p-value = 0.002).

## Conclusions

We report a new haplotype which increases susceptibility to type 1 diabetes among Italian children and which is associated with early clinical disease onset. Given the central role attributed to genetic factors in the pathogenesis of T1DM and to the II class HLA genes, this new haplotype ought to be recognized as a risk factor and included in tests routinely carried out to identify patients with a genetic predisposition to type I diabetes in Italy. These findings could have practical implications in research and prevention programs.

## Introduction

Type 1 Diabetes Mellitus (T1DM) is a chronic, autoimmune condition characterized by β-cell destruction that leads to insulin deficiency. T1DM is a polygenic disorder influenced by environmental factors including viral and bacterial infections, dietary factors and deficiency of vitamins and nutrients [1]. Genes of the major histocompatibility complex (Human Leucocyte Antigen-HLA) are currently the most reliable risk markers for the development of T1DM, the DRB1, DQA1 and DQB1 genes (HLA Class II). In Caucasians, DR4-DQ8 are associated with a high risk; numerous studies have shown that haplotypes DR3/DQ2 (DRB1 * 03-DQA1 * 05-DQB1 * 02) and DR4/DQ8 (DRB1 * 04-DQA1 * 03-DQB1 * 03: 02) raise the risk for T1DM [2, 3]. It has been hypothesized that for haplotype DR4/DQ8 the risk is determined by the allelic form of the DRB1 * 04 gene. Alleles such as DRB1 * 04: 01 and DRB1 * 04: 05 carry a high risk of disease, while others carry a moderate risk or even confer protection (e.g. DRB1 * 04: 03) [3].

In Italy, as in other countries, the most common commercial genotyping kit used to identify whether a child is susceptible to type 1 diabetes only tests for haplotypes DR3-DQ2 and DR4-DQ8. However, in the literature, there are many national-based studies in which the predisposing haplotypes are different [4–6]. El Amir et al [5] have previously published epidemiological data for Egyptian children with T1DM and demonstrated that haplotype DR4-DQ2 was also significantly increased (23.8%, OR 5.2; p value <0.0001). Manan et al [6] reported that the homozygote DQB1*0201 and DQB1*0302 and the heterozygote DQB1*0201/*0302 genotypes were all significantly higher in T1DM patients than in healthy controls (p < 0.0001).

Knowing genes implicated in T1D development risk in a given population is crucial for national prevention programs. The exact HLA types associated with disease vary among ethnic groups [2, 3].

Our study aimed to characterize a population of diabetic children followed by the Diabetology Unit of the Meyer Children's University Hospital (Florence, Italy), to assess which haplotypes are correlated with the development of T1DM. For each identified haplotype we evaluated the risk of developing the disease and possible correlations with clinical characteristics.

Greater understanding of the genes conferring a high risk for T1DM leads to better prevention programs which should be adapted to specific ethnicities and populations. Haplotype-phenotype correlations lead to a more complete understanding of the disease's pathogenicity by illuminating the role of different pathogenetic determinants.

## Materials and methods

### Study population

This is a single-center retrospective observational study that included children (aged 0–18 years) affected by T1DM and followed at the Diabetology Unit, Meyer Children's University Hospital, Italy were included. The patients were diagnosed according to the diagnostic criteria of the American Diabetes Association and ISPAD guidelines [7]. Demographical, family, clinical (comorbidities) and serological data were collected from medical report. 819 healthy adult subjects (> 18 years), registered in the Italian Bone Marrow Donor Registry, were included as controls for the case control comparison performed on HLA typing. These adult subjects were not affected by T1DM that represents an exclusion criterium for enrolling in the Italian Bone Marrow Donor Registry.

The observational period was from September 2010 to December 2018. The study was approved by the pediatric ethics committee of Meyer Children's Hospital (ID number: 162/2021 "*ABCD study*"). All results have been anonymized.

### Pancreatic autoantibody assay

GAD autoantibody (Ab), IA-2 Ab and IAA Ab were determined using a commercially available RIA kit (GAD-Ab Cosmic, Cosmic, Tokyo, Japan), as described previously [8]. Patients were judged as being positive, in accordance with previous reports, with:

- GAD Ab titers of $\geq$ 5 U/mL
- IA-2 Ab titers of $\geq$ 0.75 U/mL
- IAA Ab titers of $\geq$ 1.50 U/mL

### HLA class II typing

To determine the risk of T1DM, "DIABEGEN I˚ and II˚ step" tests of the "Eurospital kit" were used. The kit was used on DNA extracted from peripheral blood by amplification in Real Time PCR. Each sample was tested with different Real-time PCR reactions. The first step identifies the alleles predisposing to T1DM (DQA1*05/DQB1*02-DR3; DQA1*03/DQB1*03:02-DR4) and allows to detect the protective allele DQB1*06:02 and the homozygosis or heterozygosis status of DRB1*04. If we identified DRB1 * 04 heterozygosity, the second step (DiabeGen II˚ step) allows typed in high resolution (4 digits) allele DRB1 * 04 and then determine degree risk or protection. In the presence of homozygosity, the "Micro SSP Allele Specific HLA Class II DNA Typing Tray-DRB1 * 04" kit from ONE LAMBDA was used, which is based on a PCR SSP (Sequence Specific Primers) technique. This step is necessary to determine risk stratification or possible protection. According to literature [2–4] and technical instructions of used Eurospital kit "DiabeGen I˚ and II˚ step" we considered the DR-DQ haplotypes conferring the highest risk are DRB1*04-DQA1*03-DQB1*0302 (DR4-DQ8) and DRB1*03-DQA1*0501-DQB1*0201 (DR3-DQ2), the risk is much higher for the heterozygote formed by these two haplotypes, either of the homozygotes confer an intermediate risk and while the DRB1 alleles * 04: 03 and DRB1 * 04: 06 confer protection.

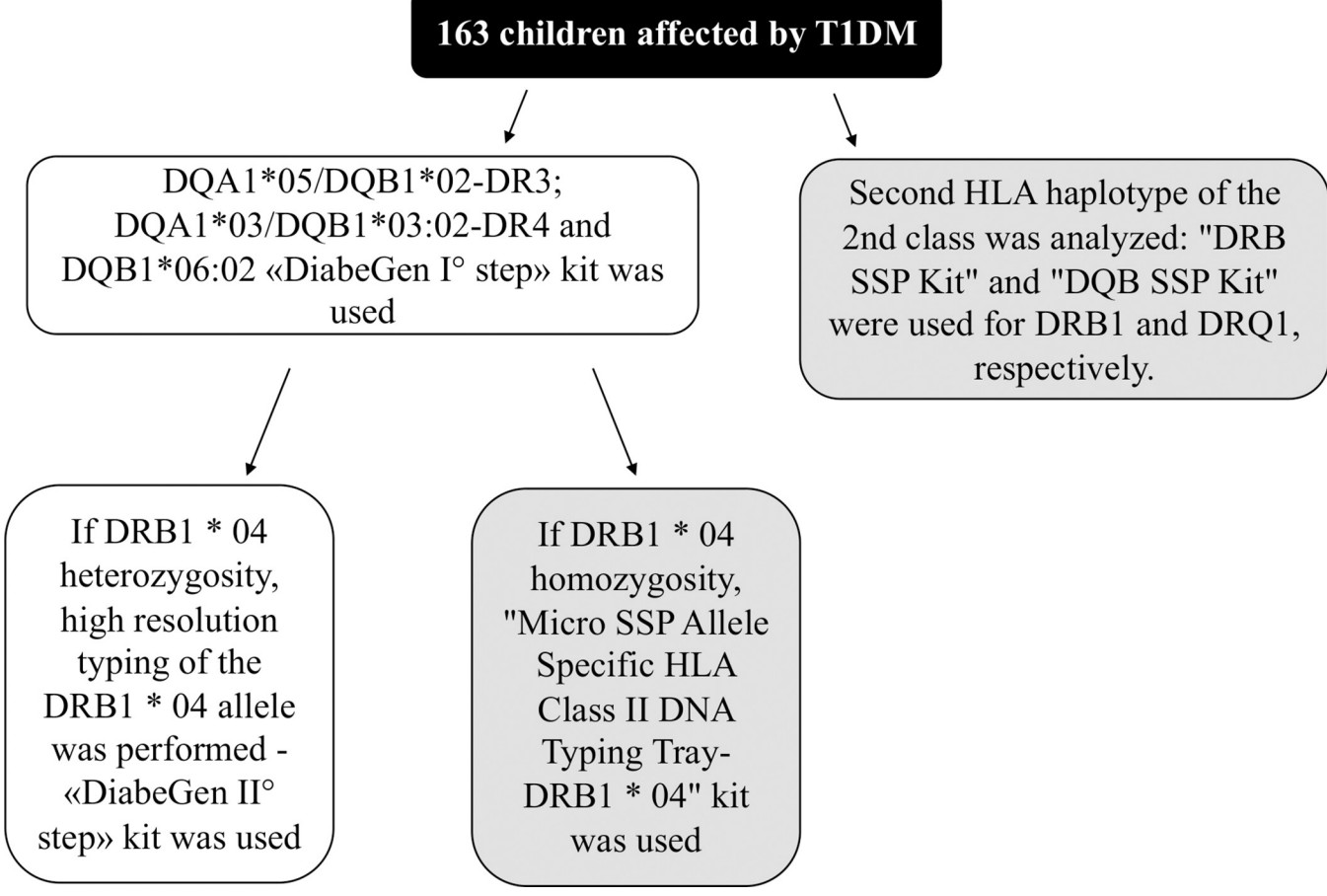

**Fig 1. Flow chart of performed HLA class II typing in T1DM population.** In white the routinary steps based on "Eurospital Diabegen I and II step"; in gray the added specifc study-steps based on PCR SSP (Sequence Specific Primers) technique.

For each patient included in the study, the second HLA haplotype of the 2nd class was studied, using the PCR SSP technique. To determine the DRB1 locus, the "DRB SSP Kit" was used and for the locus DQB1, the "DQB SSP Kit" was used, both produced by the company BIO RAD (Fig 1).

## Statistics

Data were analyzed using the JavaStat statistical program. The Chi-Square Test or the Fisher Exact Test were used when necessary and the odds ratio with confidence intervals was calculated when appropriate. Continuous variables were expressed as mean, median and standard deviation when appropriate. Student's t test was used to compare two study groups. Values of $p < 0.05$ were considered statistically significant.

## Results

### HLA class II allele frequency in T1DM patients vs healthy controls

165 children with T1DM were enrolled in the study which were divided in T1DM susceptible patients, i.e. all patients who had at least one of the haplotypes known to confer susceptibility (DR3DQ2 or DR4DQ8) and not susceptible patients, i.e. those who did not have these

haplotypes or who had protective alleles. This subdivision was made based on the results of the "Diabegen I˚ and II˚step" tests and identified 126 susceptible and 39 non-susceptible patients (39/165, 23,6%). We were able to perform HLA high definition in 163 patients: 42 patients (25.8%) had the DR3 allele in combination with the DR4 allele, 39 (23.9%) the DR3 allele with an X, 33 (20.2%) the DR4 allele with an X allele, 21 (12.9%) patients had two copies of DR3 and 4 (2.4%) patients had two copies of DR4 (homozygous patients), 24 (14.7%) had neither the DR3 allele nor the DR4 allele, but different alleles that they were identified as the X / X haplotype.

We compare these data with control population that included 819 healthy subjects. In the study population the DR3DQ2 haplotype was present 123 times compared to 165 in the control population, the DR4DQ8 haplotype was present 70 times compared to 72 in the controls, DR4DQ2 was present 10 times in patients and 4 in controls, and DR4DQX with $X \neq 8$ and $X \neq 2$ was present 3 times in patients compared to 34 times in controls (Table 1, Fig 2A). In homozygous subjects 2 copies were considered for each haplotype.

For the DR4 allele high-definition typing was performed to assess risk stratification. For each of the 83 DR4 alleles present in the study population, high-definition typing was carried out, and the frequency of each individual allele was compared to the control population.

The allele associated with the highest risk for T1DM in the study populations was DRB1 * 04: 05 (p 9.87E-21), while the haplotypes DRB1 * 04: 02 (p: 5.81E-08), DRB1 * 04: 01 (p: 5.50E-07) and DRB1 * 04: 04 (p: 1.91E-04) followed in descending order of frequency (Table 2, Fig 2B).

The haplotype DRB1 * 15: 01-DQA1 * 01-DQB2 * 06: 02 was absent in our population of patients with T1MD while present in 74 copies in the control population, including 2 copies for homozygous subjects.

In the study population we identified other alleles with less frequency (DRB1 * 04: 03 DRB1 * 04: 07 and DRB1 * 04: 17) but the low sample size does not allow us to define their protective function with statistical certainty.

All details were reported in S1 Table.

## Demographical, clinical and serological data

The group of T1DM with a haplotype no-DR4DQ2 (DR4DQ2-, 155 children) was constituted by 72 females (46.4%), the DR4DQ2+ group (8 children) include 3 females (37.5%) and 5

**Table 1. Comparison of haplotypes between the population with T1DM and healthy controls.**

|  | T1DM patientsn = 163 |  | Healthy controls n = 819 |  | OR | 95% CI | p-value |
|---|---|---|---|---|---|---|---|
|  | n | % | n | % |  |  |  |
| DR3DQ2 | 123 | 75.5% | 165 | 20.1% | 12.188 | 8.207 - 18.101 | **1.49E-45** |
| DR4DQ8 | 70 | 42.9% | 72 | 8.8% | 7.809 | 5.270 - 11.572 | **1.01E-29** |
| DR4DQ2 | 10 | 6.1% | 4 | 0.5% | 13.317 | 4.124 - 43.007 | **2.80E-08** |
| DR4DQX (X $\neq$ 8 e X$\neq$2) | 3 | 1.8% | 34 | 4.1% | 0.433 | 0.131 - 1.427 | 0.183 |

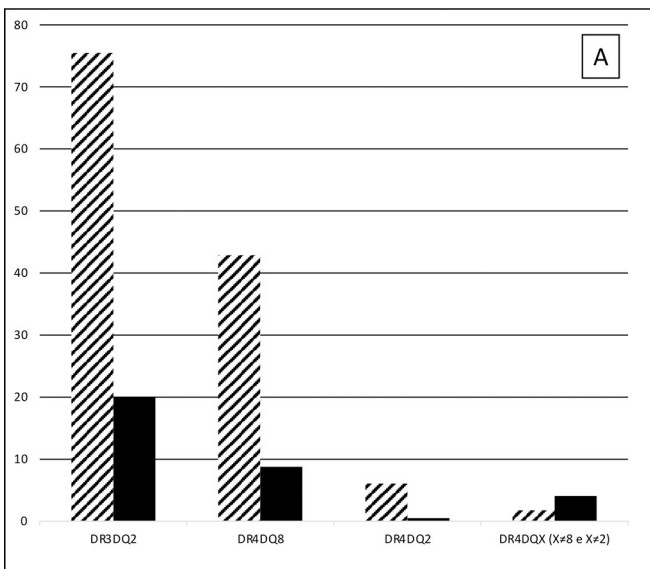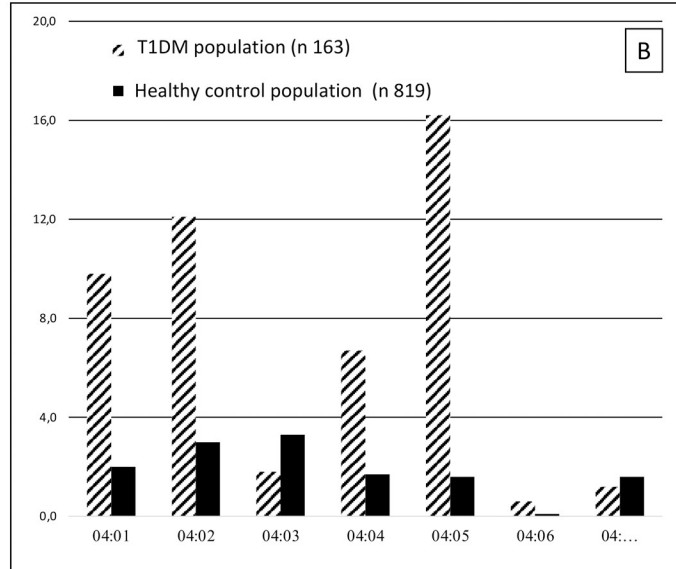

**Fig 2.** A) Comparison between the haplotypes of the T1DM population and those of the control population. B) Comparison of the DRB1 * 04 allelic variant of the study and control populations.

males (p 0.727), for the ethnicity 8/155 (5.1%) and 1/8 (12.5%) had no-Caucasian origin, in DR4DQ2- group and DR4DQ2+ group, respectively (p 0.371). Family histories for T1DM were investigated for each patient: 12/163 (7.4%) had a positive family history of which more than half (7/12) had a father with diabetes. Regarding DR4DQ2+ group 1 patient had family history positive (1/8, 12.5%), with the mother affected by T1DM. The global mean age of clinical T1DM onset was 9.4 years ± 4.2 years. For DR4DQ2+ group the mean age at diagnosis was 5.6 years ± 3.6 years, significantly lower than the mean age at diagnosis of 9 years ± 7 years for the remaining cohort (DR4/DQ2-) of 155 children with type 1 diabetes (Student's t test, p-value = 0.002). More specifically, the mean age of children with DR3DQ2 was 8.96 ± 4 years, the mean age of children with DR4DQ8 was 11.42 ± 4 years.

Regarding the association with other autoimmune phenomena associated with T1DM 47 subjects (47/163, 28.8%) had at least another autoimmune comorbidity associated with T1DM: 22 presented celiac disease (22/163, 13.3%), 22 autoimmune thyroiditis (22/163, 13.3%), 2 subject celiac disease and autoimmune thyroiditis (2/163, 1.2%) and 1 subject psoriasis (1/163,

**Table 2. Comparison of the DRB1 * 04 allelic variants of the study and control populations.** The allele variants with significant p values are in bold.

| | T1DM patients | | Healthy controls | | OR | 95% CI | p-value |
|---|---|---|---|---|---|---|---|
| | n = 163 | | n = 819 | | | | |
| | n | % | n | % | | | |
| **04:01** | 16 | 9.8% | 17 | 2% | 5.135 | 2.537–10.392 | **5.50E-07** |
| **04:02** | 21 | 12.9% | 25 | 3% | 4.697 | 2.560–8.619 | **5.81E-08** |
| **04:03** | 3 | 1.8% | 27 | 3.3% | 0.550 | 0.165–1.835 | 0.456 |
| **04:04** | 11 | 6.7% | 14 | 1.7% | 4.161 | 1.854–9.340 | **1.91E-04** |
| **04:05** | 29 | 17.8% | 13 | 1.6% | 13.418 | 6.803–26.466 | **9.87E-21** |
| **04:06** | 1 | 0.6% | 1 | 0.1% | 5.049 | 0.314–81.141 | 0.305 |
| **04: 07** | 1 | 0.6% | 6 | 0.7% | 0.836 | 0.100–6.994 | 1 |
| **04:17** | 1 | 0.6% | 7 | 0.85% | 0.716 | 0.088–5.859 | 1 |

0.6%). In 72 subjects (72/163, 44.2%) the other autoimmune comorbidity was diagnosed during follow up, after clinical disease onset, and in 30 subjects (30/163, 18.6%) the diagnosis of autoimmune comorbidity occurred before that of T1DM. For the rest of population these data were missed. The DR3/DQ2 haplotype present the higher frequency (30/47, 63.8%) in this population of subjects with another autoimmune comorbidity associated with T1DM. In the DR4/DQ2 + population 2 patients also had associated comorbidity, 1 with celiac disease, the other with 1 thyroiditis (2/8, 25%). No statistical difference was revealed between subjects DR4DQ2+ and subjects DR4DQ2- in terms of autoimmune comorbidity association frequency (p >0.999). Lastly, for 160 patients (160/165, 97%) we defined the presence of anti-pancreatic beta cell autoimmunity by evaluating IA2 (Tyrosine Phosphatase–Related Islet Antigen 2 autoantibodies), IAA (anti insulin antibodies) and anti-GAD (anti-glutamic acid decarboxylase autoantibodies). 91.4% of patients (149/163) were positive for at least one of the three antibodies, 9 patients (9/163,5.5%) were negative, 5 patients (5/163, 3,1%) did not have known autoantibody titers at disease onset. For children affected by T1DM with a clinical onset from 1 to 5 years, IA2 antibodies were more frequently positive (77.8%), followed by anti-GAD (69.4%) and finally by anti-IAA (58.3%). Among children whose diabetes was diagnosed between 6 and 12 years of age, the most frequent positive antibody was IA2 (82.8%) followed by anti-GAD (73.4%) while anti-IAA was positive in 15 patients on 64 (23.4%). Among the 54 children diagnosed above 12 years of age, 3 had negative antibodies (5.5%) and among those with serological autoimmunity the most frequently positive antibody was anti-GAD (80.4%) followed by IA2 (70.6%); IAA was found in 10 of 51 patients (19.6%). The difference of frequency of anti-IAA in different age classes was statistically significant between "1–5 years group" and "5–11 years group" (p 0.00019, OR 5.547 CI 95% 2.182–14.104) and between "1–5 years group" and "≥12 years group" (p 0.00008 OR 5.315 CI95% 2.252–12.543) (Fig 3). We compared the presence of autoantibodies in the DR4/DQ2+ group with the remaining DR4/DQ2- population: IAA2 were associated with a slightly higher frequency in the DR4/DQ2 + group (87.5% vs. 76%) and IAA antibodies in the DR4/DQ2- cohort (55.9% vs. 37.5%), but neither analysis reached statistical significance. Insufficient data are available to compare anti-GAD between the two groups.

Full details are given in S1 Table.

## Discussion

We conducted a retrospective study on 165 children diagnosed with T1DM. All children in this study population were diagnosed with diabetes according to the ADA (American Diabetes Association) criteria and ISPAD guidelines [4, 7] and are followed by the Diabetes Unit at Meyer Children's University Hospital. This study focused on high resolution evaluation of haplotype in subjects with T1DM and according to the diagnostic kit currently used in Italy, not all of them were defined as "at risk for T1DM". In this study one quarter of subjects not presented known high risk haplotypes DR3DQ2 and DR4DQ8. The high-resolution analysis of these no-susceptible subjects allows us to identify 8 patients (4.9%) with haplotype DR4/DQ2. This haplotype confers a greater susceptibility for the development of T1DM, increasing the risk by about 10 times, suggesting an important association between the specific haplotype and T1DM. The DRB1 * 04: 05 allele has the highest risk of T1DM, followed in descending order by the allele DRB1 * 04: 01, DRB1 * 04: 02 and by DRB1 * 04: 04. This study shows an association between T1DM and a new HLA susceptibility haplotype, DR4DQ2. The haplotype has been previously highlighted in other countries, such as Egypt, Saudi Arabia, Macedonia, Ethiopia and Cyprus [5, 6, 9–11]. But this is the first study that demonstrated this association in Italy. These results strongly suggest that the HLA DR4DQ2 haplotype, and in particular the

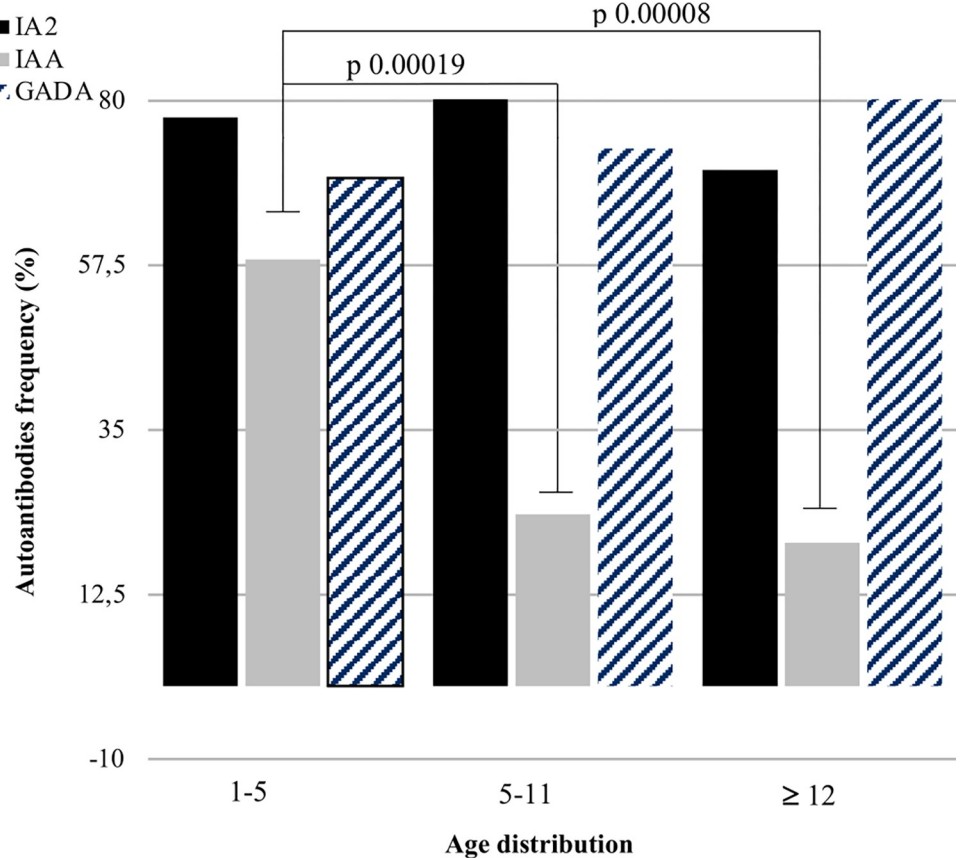

**Fig 3. Autoantibodies anti-IA2, anti-IAA e anti-GAD frequencies: Age distribution.**

HLA DRB1 * 04: 05-DQA1 * 03-DQB2 * 02 allelic form, predisposes carriers to T1DM with a risk between that of the DR3DQ2 haplotype and the haplotype DR4DQ8. It appears crucial to determine the risk factors associated with HLA haplotypes in specific populations to ameliorate screening performance and diagnostic accuracy. Furthermore, we performed a sub-analysis of available demographic and serological and clinical data between the two group DR4DQ2- and DR4DQ2+, in order to better characterize this subgroup. We demonstrated DR4DQ2+ haplotype is associated with early clinical onset of diabetes in comparison with other haplotypes, confirmed the reported results of a study carried out in Central Europe, where HLA-DQA1*03-DQB1*02 haplotype was identified more rarely but associated with early onset [12].

We also investigated the association with other autoimmune comorbidities, confirming celiac disease or autoimmune thyroiditis (27.3%) had a higher association frequency. e observed that most children with diabetes and celiac disease had the DR3DQ2 haplotype, which is known to have a strong association with celiac disease [13]. Instead, among children with diabetes and autoimmune thyroiditis, the DR3DQ2 haplotype occurred with the same frequency as the DRB1 * 04 allele suggesting a lower association with thyroiditis. In subgroup of subjects with DR4DQ2 haplotype no significant difference were identified in autoimmune comorbidities association. The autoimmune comorbidities more often diagnosed after determination of T1DM, confirming that their screening during follow up of subjects affected by T1DM is essential, accordingly with ISPAD guidelines.

For family case accordingly with the literature the risk of diabetes is higher if the diabetic parent is the father than the mother or brother or sister [14]. This phenomenon has generated both genetic and non-genetic hypotheses but the mechanism of transmission of T1DM remains unclear [15]. The small size of DR4DQ2 group does not allow any conclusion about familiarity association and specific haplotype. The global sensitivity of autoantibodies positivity (at least one of three) in this study was confirmed elevated (94.4%) and the most frequently positive antibodies are IA2 and anti-GAD in all age groups of diagnosis. However, these results confirm the anti-insulin antibody (IAA) is more frequent among children with earlier diagnoses (0–5 years of age) and that they are usually the first to appear. The causes are still to be defined but it is possible that IAAs are a marker of rapid beta cell destruction and therefore more frequently present in patients who present a rapid progression of destructive insulitis. Supporting this hypothesis is experimental evidence that IAAs correlate with the rate of loss of cellular beta function [16, 17]. Regarding, the frequency of autoantibodies among diabetic children with DR4DQ2 haplotype was found to be similar to that of the rest of the study population; this feature differs from that of the Egyptian population described by El-Amir and colleagues, where the haplotype DR4DQ2 was recognized as a risk for diabetes but associated with a low frequency of all antibodies [5]. Moreover, despite the early age of onset of T1DM for subjects with DR4DQ2, there was a lower prevalence of the anti-insulin antibody (IAA) in children with this haplotype than in other children with early diagnoses without the haplotype DR4DQ2. This antibody is the first to appear in children under the age of 5 and is associated with an early diagnosis and we expected a higher prevalence in DR4DQ2+ group. Based on this consideration together with the early onset of T1DM in DR4DQ2+ group led us to speculate that could be exist different pathogenetic mechanisms in children with DR4DQ2 haplotype. The small size of our cohort does not allow us to draw significant conclusions, but these finding suggests that larger multicenter studies should be conducted.

## Conclusion

This study confirms in Italian population the strong association between T1DM and the DR3DQ2 and DR4DQ8 haplotypes, already known to confer high risk, and confirmed the protective role of the haplotype DRB1 * 15: 01-DQA1 * 03-DQB1 * 06: 02. Furthermore, this study identified the DR4DQ2 haplotype as a risk factor for the development of T1DM in our region, confirming some studies conducted in other Mediterranean countries. In addition, DR4DQ2 haplotype appears to be related to an earlier age of symptom onset. We believe that it is important to consider an individual's risk of developing T1DM in relation to their ethnic-geographical background, and to confirm or identify new risk conferring haplotypes in order to design more accurate prevention programs and for research purposes.

## Supporting information

**S1 Table. All demographic, clinical and serological details for 163 subjects affected by T1DM with HLA high-resolution.**
(DOCX)

## Author Contributions

**Conceptualization:** Silvia Ricci, Sonia Toni, Chiara Azzari.

**Data curation:** Silvia Ricci, Francesca Perugia, Sonia Toni.

**Formal analysis:** Silvia Ricci, Francesca Perugia, Sonia Toni.

**Methodology:** Silvia Ricci, Giancarlo Perferi.

**Supervision:** Silvia Ricci, Sonia Toni, Chiara Azzari.

**Validation:** Silvia Ricci, Lorenzo Lodi, Francesco Pegoraro, Mattia Giovannini, Giancarlo Perferi, Sonia Toni, Chiara Azzari.

**Visualization:** Silvia Ricci, Barbara Piccini, Lorenzo Lodi, Francesco Pegoraro, Mattia Giovannini, Giancarlo Perferi, Sonia Toni, Chiara Azzari.

**Writing – original draft:** Silvia Ricci, Francesca Perugia.

**Writing – review & editing:** Silvia Ricci, Barbara Piccini, Lorenzo Lodi, Francesco Pegoraro, Mattia Giovannini, Giovanni Rombolà, Giancarlo Perferi, Sonia Toni, Chiara Azzari.

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
