## [Decision Letter · Decision Letter 0]

12 Jul 2022

PONE-D-22-02849DR4/DQ2 haplotype confers susceptibility to T1DM with early clinical disease onset: a retrospective analysis in a tertiary-care hospital in ItalyPLOS ONE

Dear Dr. Ricci,

Thank you for submitting your manuscript to PLOS ONE. After careful consideration, we feel that it has merit but does not fully meet PLOS ONE’s publication criteria as it currently stands. Therefore, we invite you to submit a revised version of the manuscript that addresses the points raised during the review process.Please ensure that your decision is justified on PLOS ONE’s publication criteria and not, for example, on novelty or perceived impact.

We look forward to receiving your revised manuscript.

Kind regards,

Giuseppe Novelli

Academic Editor

PLOS ONE

4. Figures/tables should not be in both the manuscript, and as separate files.

Reviewers' comments:

Reviewer's Responses to Questions

**Comments to the Author**

1. Is the manuscript technically sound, and do the data support the conclusions?

Reviewer #1: Yes

Reviewer #2: Yes

2. Has the statistical analysis been performed appropriately and rigorously? 

Reviewer #1: Yes

Reviewer #2: No

3. Have the authors made all data underlying the findings in their manuscript fully available?

Reviewer #1: Yes

Reviewer #2: Yes

4. Is the manuscript presented in an intelligible fashion and written in standard English?

Reviewer #1: Yes

Reviewer #2: No

5. Review Comments to the Author

Reviewer #1: The authors present a hospital cohort of early clinical onset T1DM patients analyzed for genetic factors linked to HLA haplotypes with the aim of identifying specific regional haplotypes in addition to the prevalent ones. The cohort is not large and the authors correctly declare the need for further validations, the analysis is conducted with validated methodologies, the results show the presence of the DR4 / DQ2 haplotype significant for susceptibility to T1DM and not present in the Italian standard protocols. To support and clarity of the results presented and the consequent conclusions, the presentation of the demographic data of the cohort and DR4 / DQ2 patients is necessary in order to verify the clinical and geographic specificity of the variant.

Reviewer #2: 1) The abstract is not too much focused on the topic. Abstract: delete “factors” from the second line, there is a repeat. In materials and methods there is a “s”. “Familiarity” is a false friend; family history sounds better. It is better to say medical data or report than “medical files”. The % in the parentheses missed the number. The part with the comorbidities in the abstract is useless there. You should insert instead the p values of the HLA haplotypes found in DM1 population.

2) Introduction: You should underline the p values of the other studies done to evaluate the HLA-T1D correlation.

3) Materials and methods: not medical files, but medical data. You should specify how many people the control group consists of, average age, and if you are sure that we do not have the dm1. In the Italian donor register, do participants specify whether they are suffering from autoimmune diseases or is this an exclusion criterion?

4) HLA Class II typing: explain the methods of analysis of the different HLA in order or with a flowchart or table. Explaining here which HLAs give increased, intermediate, and low risk is not indicated. To put this data in the introduction, specifying the sources and p value. The figure 1 is good, but with p values.

5) Results: “5 patients did not have known autoantibody at disease onset”. Did you test them now? “the difference of frequency of anti IAA ………(Figure 1)”; what statistical test was used? Chi Quadro for 3 different variables? Regarding the onset, you should correlate them with HLA (e.g. onset 10-15 ys � HLA XXX)

6) HLA class II allele frequency in T1D: in the second line there is “and”.

7) You should put all the data of HLA correlations in the tables in the same table.

8) DR4/DQ2 haplotypes: “study population” is not accurate. You should use “the population with T1D”. There is also a repetition. “we compared -………DR4/DQ2- cohort” � there is or not the difference of this IA2 antybodies?

9) HLA and comorbidities: you should put the data in a table. You should correlate them with the HLA Haplotype.

10) References: you should correct the references, as not all written in Vancouver (after the authors does not go the point, the pages should be shortened e.g. 24-26  24-6)

11) You should rewrite the data in a better way, more clear and short, without repeats.

12) In the figures, you should use them in English, not in Italian.

6. PLOS authors have the option to publish the peer review history of their article (what does this mean?). If published, this will include your full peer review and any attached files.

Reviewer #1: No

Reviewer #2: No

---

## [Author Response · Author response to Decision Letter 0]

24 Sep 2022

Dear reviewers, thank you for taking the time to review our article. In the revised manuscript we tried to follow all your suggestions and advice. As you will see, we have substantially changed the entire article. I have added the requested table, if you agree in the Supplementary Materials and I have created a third figure to explain the methods session. 

I hope the result is better and can be accepted for publication, however, please do not hesitate if there is any other criticism and advice. 

Thank you

Silvia Ricci

Review Comments to the Author

Reviewer #1: The authors present a hospital cohort of early clinical onset T1DM patients analyzed for genetic factors linked to HLA haplotypes with the aim of identifying specific regional haplotypes in addition to the prevalent ones. The cohort is not large and the authors correctly declare the need for further validations, the analysis is conducted with validated methodologies, the results show the presence of the DR4 / DQ2 haplotype significant for susceptibility to T1DM and not present in the Italian standard protocols. To support and clarity of the results presented and the consequent conclusions, the presentation of the demographic data of the cohort and DR4 / DQ2 patients is necessary in order to verify the clinical and geographic specificity of the variant.

Thank you, I have included the table with all available details as requested.

Reviewer #2: 1) The abstract is not too much focused on the topic. Abstract: delete “factors” from the second line, there is a repeat. In materials and methods there is a “s”. “Familiarity” is a false friend; family history sounds better. It is better to say medical data or report than “medical files”. The % in the parentheses missed the number. The part with the comorbidities in the abstract is useless there. You should insert instead the p values of the HLA haplotypes found in DM1 population.

Author: Thank you very much for your advice which I followed in revising the abstract which should now be more concise and aimed at making the reader understand the the meaning of our work.

2) Introduction: You should underline the p values of the other studies done to evaluate the HLA-T1D correlation.

Author: Ok, I have added these data, as suggested. 

3) Materials and methods: not medical files, but medical data. You should specify how many people the control group consists of, average age, and if you are sure that we do not have the dm1. In the Italian donor register, do participants specify whether they are suffering from autoimmune diseases or is this an exclusion criterion?

Author:Thank you for the questions. We have better specified in the methods session the characterization of the control group by including the number of subjects and specifying that they are all subjects over 18 years of age without T1MD because this is an exclusion criterion for enrolling a subject in the marrow donor registry.

4) HLA Class II typing: explain the methods of analysis of the different HLA in order or with a flowchart or table. Explaining here which HLAs give increased, intermediate, and low risk is not indicated. To put this data in the introduction, specifying the sources and p value. The figure 1 is good, but with p values.

Author: Ok, thank you. I have prepared a new figure I with the flowchart of method. I have explain wich HLAs are associated to develop T1DM with increased/intermediate and low risk. I have added p values in Figure 2 (ex figure 1), as you suggested. Thank you.

5) Results: “5 patients did not have known autoantibody at disease onset”. Did you test them now? “the difference of frequency of anti IAA ………(Figure 1)”; what statistical test was used? Chi Quadro for 3 different variables? Regarding the onset, you should correlate them with HLA (e.g. onset 10-15 ys � HLA XXX)

Author:I am sorry but we have not these more data regarding autoantibodies during disease. The difference of frequency of IAA are analyzed by X2 test between “1-5 ys” group vs “5-11 ys” group and between “1-5 ys” group vs “>12 ys” group (Figure 2). I have added mean age and DS for the three more frequent haplotypes. 

6) HLA class II allele frequency in T1D: in the second line there is “and”.

Author:Ok, done.

7) You should put all the data of HLA correlations in the tables in the same table.

Author:Ok I created a table with all available details for 163 subjects. 

8) DR4/DQ2 haplotypes: “study population” is not accurate. You should use “the population with T1D”. There is also a repetition. “we compared -………DR4/DQ2- cohort” � there is or not the difference of this IA2 antybodies?

Author:Thank you, I agree, that part it was confusing. We have rephrased the sentence. I hope it is simpler to read. 

9) HLA and comorbidities: you should put the data in a table. You should correlate them with the HLA Haplotype.

Author:Ok I created a table with all available details for 163 subjects. 

10) References: you should correct the references, as not all written in Vancouver (after the authors does not go the point, the pages should be shortened e.g. 24-26  24-6)

Author:I have revised and re-styling the reference session (I copied all references directly by PubMed). 

11) You should rewrite the data in a better way, more clear and short, without repeats.

Author:I think I have substantially reformulated the presentation of results and discussion, following your advice, and I think the work has improved. I look forward to your feedback and any other comments or corrections.

12) In the figures, you should use them in English, not in Italian.

Author:I am so sorry for this mistake. I have internally presented these data as power point in my group in Florence and I have probably exchanged the files delle figure.

---

## [Editor Report · Decision Letter 1]

17 Oct 2022

DR4/DQ2 haplotype confers susceptibility to T1DM with early clinical disease onset: a retrospective analysis in a tertiary-care hospital in Italy

PONE-D-22-02849R1

Dear Dr. Ricci,

We’re pleased to inform you that your manuscript has been judged scientifically suitable for publication and will be formally accepted for publication once it meets all outstanding technical requirements.

Kind regards,

Giuseppe Novelli

Academic Editor

PLOS ONE